# Anatomical factors associated with the lingual fracture pattern in sagittal split ramus osteotomy: A case-control study

Thalles Moreira Suassuna[1], Elenisa Glaucia Ferreira dos Santos[1],
Sérgio Murilo Cordeiro de Melo Filho[2], Carla Cecília Lira Pereira de Castro[2],
Tatiane Fonseca Faro[3], Fernanda Souto Maior dos Santos Araújo[4],
Allan Vinícius Martins-de-Barros[1,3,5], Fábio Andrey da Costa Araújo[1,2]*

1 University of Pernambuco, Postgraduate Program in Dentistry, Recife, Brazil, 2 Department of Oral and Maxillofacial Surgery, School of Dentistry, University of Pernambuco, Recife, Brazil, 3 Hospital dos Servidores do Estado de Pernambuco, Recife, Brazil, 4 School of Dentistry, University of Pernambuco, Recife, Brazil, 5 University of Pernambuco, Postgraduate program in Health and Social and Environmental Development, Campus Arcoverde, Arcoverde, Pernambuco, Brazil

* fabio.andrey@upe.br

## Abstract

Unfavorable fractures are among the most challenging complications in sagittal split ramus osteotomy (SSRO), potentially increasing surgical morbidity and compromising postoperative outcomes. The preoperative identification of anatomical risk factors through imaging can enhance surgical planning and prevent such events. This study aims to investigate the anatomical factors associated with lingual fracture patterns in SSRO using multislice computed tomography. This retrospective case-control study included 180 mandibular rami from patients who underwent SSRO at a Clinical Research Center for Oral and Maxillofacial Surgery. Fractures were classified according to Plooij (2009), with Types 3 and 4 grouped as cases and Types 1 and 2 as controls. Linear measurements of mandibular ramus thickness, the distance between the mandibular canal and the buccal cortical bone, as well as parameters related to the mandibular lingula were analyzed. The case group showed significantly thinner mandibular bone ($p < 0.001$) and a shorter canal-cortical distance ($p = 0.013$), suggesting a direct association between these anatomical variables and unfavorable fracture patterns. Bonferroni post hoc analysis revealed no significant difference between fracture patterns Type 3 and Type 4 ($p = 1.000$), supporting their grouping in a single analytical category. The presence of third molars was not significantly associated with fracture patterns ($p > 0.05$). These results underscore the importance of anatomical parameters in predicting the risk of unfavorable fractures. Specifically, reduced bone thickness and proximity of the mandibular canal play a crucial role in the occurrence of unfavorable SSRO fractures. Preoperative evaluation using computed tomography is essential to optimize surgical planning and minimize complications. However, given the limitations of retrospective designs potential biases are acknowledged, and

**Data availability statement:** All relevant data are within the manuscript and its Supporting Information files.

**Funding:** The author(s) received no specific funding for this work.

**Competing interests:** The authors have declared that no competing interests exist.

further prospective studies are needed to confirm these findings and improve risk assessment in SSRO.

## Introduction

Sagittal split ramus osteotomy (SSRO) is widely used in orthognathic surgery to correct dentofacial deformities such as prognathism, retrognathism, and mandibular asymmetries [1–4]. Initially described by Trauner and Obwegeser in 1957 and later modified by Dal Pont, the technique has been refined over the years, but the splitting step remains a challenging aspect of orthognathic surgery [5,6]. SSRO still presents some limitations, including difficulties in controlling the fracture pattern on the lingual surface, which may lead to unfavorable fractures (UF).

Plooij et al. (2009) [7] classified mandibular fracture patterns during the bone split in SSRO into four types: Type 1 is a vertical fracture extending toward the inferior border of the ramus; Type 2 is a horizontal fracture directed toward the posterior border of the ramus; Type 3 is a vertical fracture passing through the mandibular canal toward the inferior border; and Type 4 is an unfavorable fracture involving the coronoid process and/or subcondylar region. The last two types may represent up to 20% of cases and are classified as non-conventional fracture patterns, thus increasing the risk of complications during orthognathic surgery [7–11].

The main complications associated with these fracture patterns include increased surgical trauma, infection, temporomandibular joint complications, and bone non-union [7,12,13], making orthognathic surgery more challenging and associated with higher morbidity for the patient.

Studies suggest that anatomical factors may influence the risk of UF, such as the thickness of the cortical bone, fusion between buccal and lingual cortices, and the presence of third molars [7,14–16]. Jiang (2021) demonstrated that a shorter distance between the mandibular canal and the buccal cortex is associated with a higher risk of unfavorable fracture.

Understanding the anatomical characteristics that influence the fracture path—especially in the lingual region of the mandible—may contribute to more predictable and individualized UF risk assessment, reduced complications, and improved SSRO technique. Therefore, the aim of this study was to evaluate mandibular anatomical factors that may influence the fracture pattern on the lingual aspect of the mandibular ramus during SSRO.

## Methodology

### Study design and ethical considerations

This is a retrospective case-control study conducted at the Clinical Research Center in Oral and Maxillofacial Surgery and Traumatology of the Oswaldo Cruz University Hospital, in Recife, Brazil, from December 2024 to January 2025. The study was approved by the Research Ethics Committee of the University of Pernambuco (Approval No. 6,570,319; CAAE No. 74613823.2.0000.5207), and conducted in

accordance with the Strengthening the Reporting of Observational Studies in Epidemiology (STROBE) guidelines for case-control studies [17]. All patients were informed about the risks and benefits of the study and signed an Informed Consent Form for the use of their tomographic images and the information contained in their medical records.

## Study population and eligibility criteria

The study population consisted of individuals who underwent SSRO for correction of dentofacial deformities at a single hospital center in Northeast Brazil. All surgical procedures were performed by the same experienced oral and maxillofacial surgical team using the SSRO technique described by Dal Pont and modified by Hunsuck and Epker, followed by hybrid osteosynthesis using titanium plates and screws.

Eligible participants were men and women over 18 years old who had undergone surgical treatment for dentofacial deformities between January 2022 and December 2024 and had both pre- and postoperative tomographic images archived at the institution. Individuals with a history of facial trauma, reoperation, or no available postoperative CT imaging within 30 days of surgery were excluded.

For this study, each mandibular ramus was considered an independent unit of analysis, given that the fracture pattern occurs independently in both sides of SSRO. Mandibular rami were divided into two groups based on postoperative CT scan analysis of the lingual split pattern in SSRO. Fractures were classified according to Plooij et al. (2009):

- Type 1: Vertical fracture extending toward the inferior border (True Hunsuck)
- Type 2: Horizontal fracture directed toward the posterior border of the ramus
- Type 3: Vertical fracture line through the mandibular canal toward the inferior border
- Type 4: Other or unfavorable fracture patterns

The **case group** included mandibular rami with non-conventional or unfavorable fracture patterns (Types 3 and 4), while the **control group** included rami with conventional fracture patterns (Types 1 and 2) (Fig 1).

## Sample size

For the sample size calculation, a significance level of 5% ($\alpha = 0.05$), statistical power of 85% ($1 - \beta = 0.85$), a case-to-control ratio of 1:1, and an expected prevalence of the primary exposure variable (mandibular buccal cortical thickness < 4 mm) of 36.9% among controls and 59.3% among cases were considered. Based on these parameters and using the formula for comparing two proportions in unmatched case-control studies, a minimum sample size of 90 units of analysis per group was obtained, totaling 180 mandibular rami.

## CT image analysis

To evaluate mandibular anatomical parameters, pre and immediate postoperative facial computed tomography (CT) scans were performed using a multislice scanner, with 0.7 mm slices, 0.4 mm voxel resolution, and parameters of 120 kV and 100 mAs/slice, saved in DICOM format.

Preoperative images were retrospectively analyzed using Dolphin Imaging® software (version 11.95). A single examiner, blinded to group allocation, conducted all image evaluations. The assessed parameters included linear measurements of the mandibular ramus and the presence or absence of third molars in the mandibular arch. Linear measurements were obtained using the "Measure" tool in 2D line mode in the aforementioned software.

The thickness of the mandibular ramus at the level of the lingula was measured on coronal slices, while the distance between the buccal cortex and the mandibular canal, the distance between the sigmoid notch and the inferior border of the mandibular ramus, the distance from the posterior to the anterior border of the mandibular ramus, and the linear

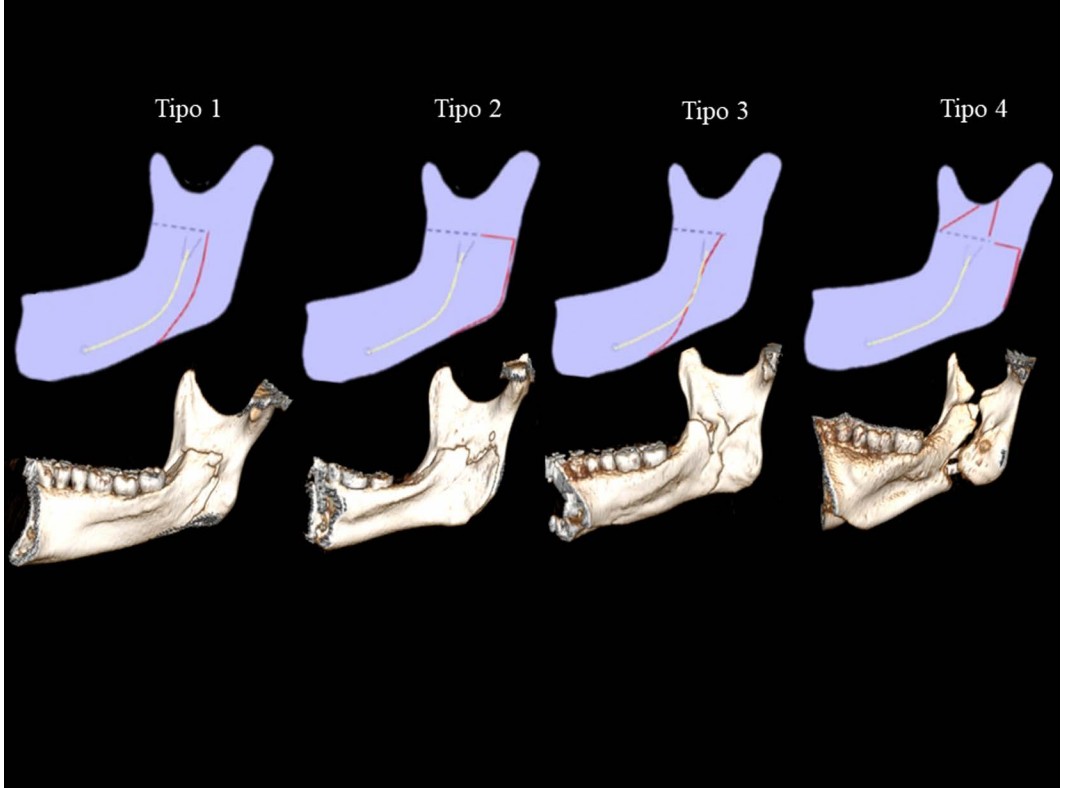

**Fig 1. Comparison between Plooij's (2009) classification and the patterns observed in the study based on 3D CT reconstructions.**

distances from the lingula to the anterior, inferior, and posterior borders of the ramus were obtained from 3D CT reconstructions. Additionally, the presence or absence of third molars was assessed using the panoramic reconstruction view (Fig 2).

(A) **Linear distance from the ML to the sigmoid notch**

(B) **Distance from the ML to the anterior border of the mandibular ramus**

(C) **Distance from the ML to the inferior border of the mandibular ramus**

(D) **Distance from the ML to the posterior border of the mandibular ramus**

(E) **Ramus height (F) Ramus length**

Postoperative CT scans were analyzed using RadiAnt DICOM Viewer® (v25.1). The 3D reconstructions were generated from axial slices through selective segmentation of the mandible to evaluate the lingual fracture pattern of the mandibular ramus (Fig 1).

**Database construction and statistical analysis**

A database was created using SPSS® software (Statistical Package for the Social Sciences), version 20.0, where descriptive and inferential statistical analyses were performed. Descriptive analysis included measures of central tendency and dispersion for quantitative variables, and absolute and relative frequencies for categorical variables.

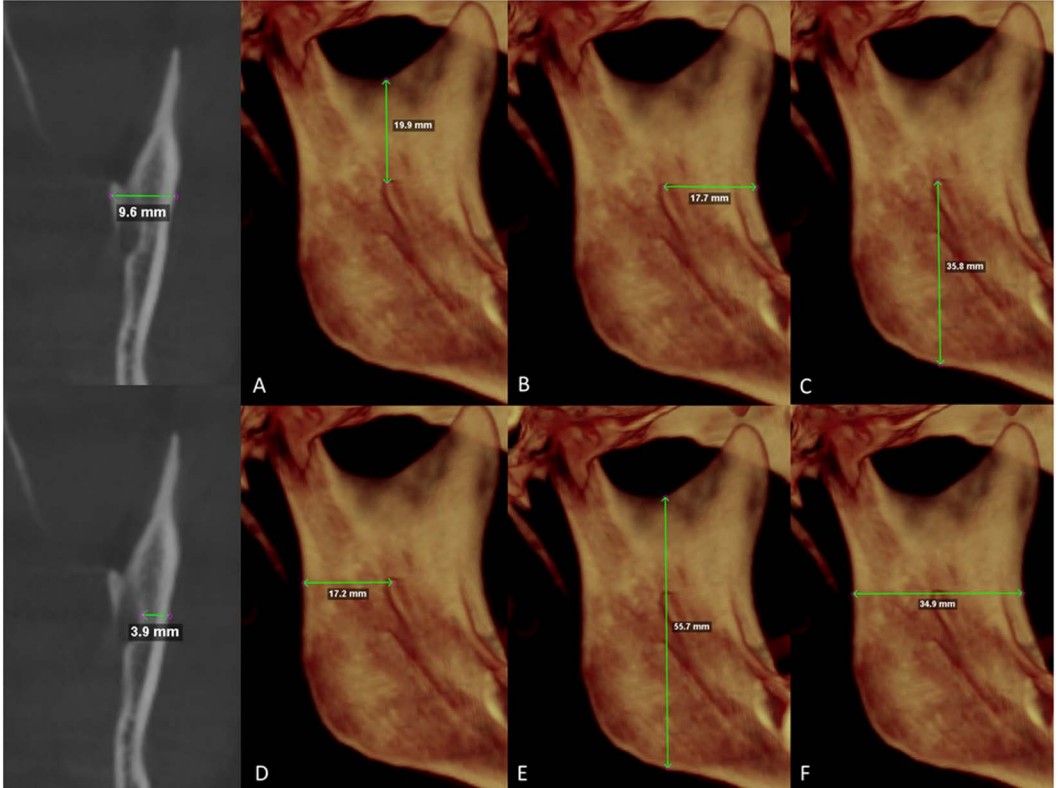

**Fig 2. Linear measurement from the buccal cortex of the mandibular ramus to the outer cortex of the mandibular canal and the ramus thickness at the level of the mandibular lingula (ML).**

Inferential statistical analysis was conducted to assess possible associations between SSRO fracture patterns and predictive variables. For quantitative predictors, the Kolmogorov–Smirnov normality test was performed, and since data followed a normal distribution, group comparisons were made using the independent samples t-test. For categorical predictors, the chi-square test with Bonferroni correction was applied. A significance level of 95% was adopted for all analyses.

## Results

Initially, 390 mandibular rami submitted to SSRO were evaluated through tomographic records from the institution's database. Of these, 177 were excluded due to the unavailability of immediate postoperative CT scans (within 30 days of surgery), and 213 were considered eligible for the study and were retrieved for analysis. After reviewing the tomographic images, 33 mandibular rami were excluded—5 due to a history of facial trauma and 28 due to previous reoperation.

The final sample consisted of 180 mandibular rami from 108 consecutive participants. The rami were equally distributed between the case group (n = 90 mandibular rami from 72 participants) with non-conventional fracture patterns (Types 3 and 4) and the control group (n = 90 mandibular rami, 71 participants) with conventional fracture patterns (Types 1 and 2) (Fig 3). Overall, 37 participants had one mandibular ramus allocated to the case group and the other to the control group.

The demographic and clinical characteristics of the sample are shown in Table 1. In the case group, 50 (47.2%) participants were female and 40 (54.1%) were male. In the control group, 56 (52.8%) were female and 34 (45.9%) male. The participants' ages ranged from 18 to 52 years, with a mean age of 27 (± 7.4) years. Regarding the type of deformity, 2 (1.1%) participants were classified as Class I, 56 (31.1%) as Class II, and 122 (67.8%) as Class III. Among the 25 (13.9%)

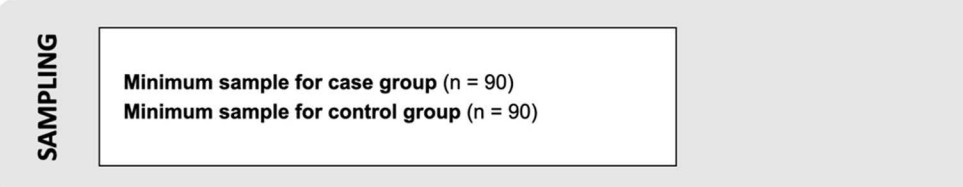

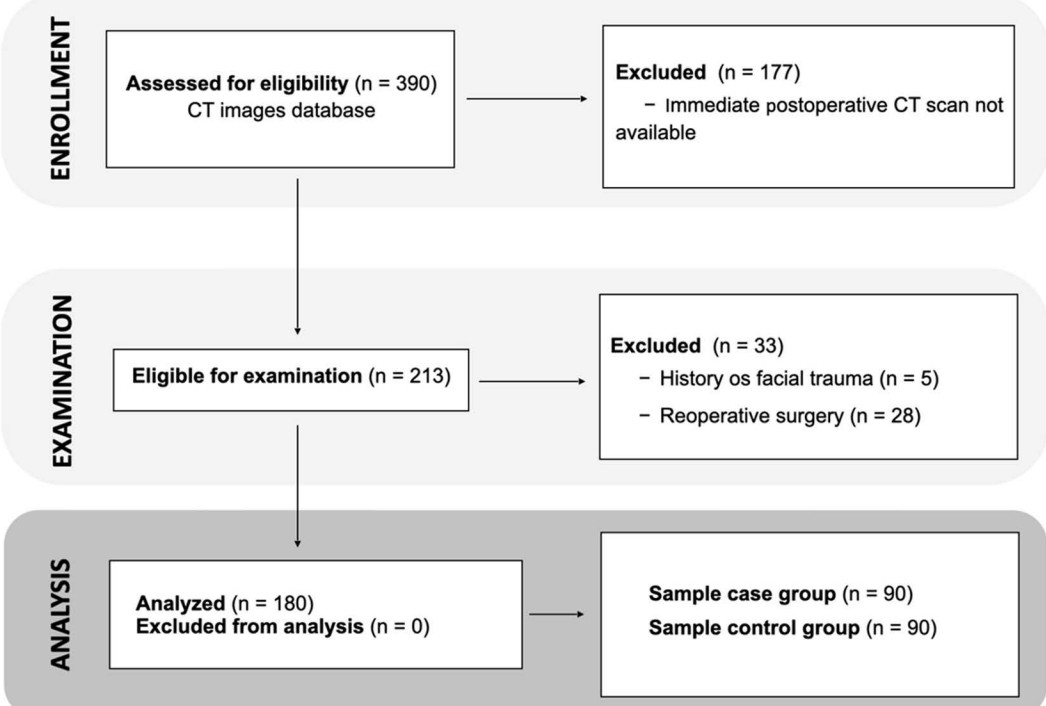

**Fig 3. Flowchart of the sample selection process.**

**Table 1. Distribution of 90 SSRO cases in each group according to sex, age, deformity type, and presence of third molars.**

| Variables | Case (n = 90) | Control (n = 90) | p-value |
|---|---|---|---|
| Sex – Female | 50 (47.2%) | 56 (52.8%) | 0.363 |
| Sex – Male | 40 (54.1%) | 34 (45.9%) | |
| Age (mean ± SD) | 27 ± 7.4 | 28.3 ± 7.4 | 0.25 |
| Age ≤ 27 years | 45 (50.0%) | 42 (47.2%) | 0.361 |
| Age > 27 years | 45 (50.0%) | 48 (54.2%) | |
| Class I | 0 (0.0%) | 2 (0.9%) | |
| Class II | 30 (53.6%) | 26 (46.4%) | 0.314 |
| Class III | 60 (49.2%) | 62 (50.8%) | |
| 3rd Molar Present | 13 (52.0%) | 12 (48.0%) | |
| 3rd Molar Absent | 77 (49.7%) | 78 (50.3%) | 0.829 |

*p < 0.05.

participants who presented with third molars preoperatively, 13 (52.0%) were in the case group. No statistically significant differences were observed between the case and control groups regarding demographic data, third molar presence, or type of deformity (p > 0.05) (Table 1).

The evaluation of mandibular anatomical factors aimed to identify potential associations with fracture patterns, and the results are presented in Table 2. The findings show that both the canal-cortical distance and mandibular ramus thickness were significantly lower in the case group. Conversely, the total length and height of the ramus did not differ significantly between groups (p = 0.063; p = 0.987). Distances between the mandibular lingula (ML) and the sigmoid notch, as well as the anterior, inferior, and posterior ramus borders, showed no significant differences between conventional and non-conventional fracture groups.

Bonferroni post hoc analysis was used to compare differences in mandibular ramus thickness between SSRO fracture patterns according to Plooij's (2009) classification. The multiple comparison revealed no statistically significant difference in ramus thickness between fracture types 3 and 4 (p = 1.000), suggesting both patterns share similar biomechanical characteristics (Table 3).

## Discussion

The occurrence of unfavorable fractures during sagittal split ramus osteotomy (SSRO) has been widely investigated in the literature with the aim of identifying anatomical risk factors that may predispose to this type of complication. The results of this study are consistent with previous findings indicating that bone thickness and the distance between the mandibular canal and buccal cortex are significantly associated with non-conventional fracture patterns [16,18].

Table 2. Analysis of mandibular anatomical factors associated with fracture pattern in the groups.

| Variables | Case (n = 90) | Control (n = 90) | p-value |
|---|---|---|---|
| Canal-cortical distance (mm) | 3.9 ± 1.0 | 4.4 ± 1.3 | 0.013* |
| Ramus thickness (mm) | 4.0 ± 1.1 | 8.7 ± 1.4 | 0.000* |
| Ramus length (mm) | 27.8 ± 3.4 | 28.8 ± 3.4 | 0.063 |
| Ramus height (mm) | 44.8 ± 6.1 | 44.8 ± 5.9 | 0.987 |
| ML–sigmoid notch (mm) | 15.1 ± 2.8 | 15.3 ± 3.2 | 0.669 |
| ML–anterior border (mm) | 15.0 ± 2.0 | 15.5 ± 2.0 | 0.085 |
| ML–inferior border (mm) | 29.6 ± 4.3 | 29.4 ± 4.0 | 0.739 |
| ML–posterior border (mm) | 12.8 ± 2.5 | 13.2 ± 1.9 | 0.219 |

*p < 0.05.

ML: Mandibular Lingula.

Table 3. Multiple comparison between fracture patterns for mandibular ramus thickness using Bonferroni post hoc test.

| Compared Patterns | Mean Difference (mm) | 95% CI Lower | 95% CI Upper | p-value | Significant |
|---|---|---|---|---|---|
| Type 1 vs. Type 2 | −0.425 | −1.035 | 0.185 | 0.959 | No |
| Type 1 vs. Type 3 | 4.569 | 4.003 | 5.135 | 0.000 | Yes |
| Type 1 vs. Type 4 | 4.781 | 3.948 | 5.614 | 0.000 | Yes |
| Type 2 vs. Type 3 | 4.994 | 4.418 | 5.569 | 0.000 | Yes |
| Type 2 vs. Type 4 | 5.206 | 4.382 | 6.029 | 0.000 | Yes |
| Type 3 vs. Type 4 | 0.212 | −0.813 | 1.236 | 1.000 | No |

*Bonferroni post hoc testing adjusts the p-value for multiple comparisons to reduce the risk of false positives. A p-value greater than 0.05 indicates no significant difference between groups.

Mandibular ramus thickness was one of the main variables associated with fracture pattern. Our data demonstrated that reduced ramus thickness was significantly more frequent in the case group, a finding consistent with previous studies showing that thinner anatomical structures along the osteotomy pathway are associated with a higher predisposition to unfavorable fractures [16,18]. Telha et al. (2023) observed that a reduction in cancellous bone thickness, as well as decreased distance between the mandibular canal and the buccal cortex in the region of the first molar, led to unfavorable fractures during surgery.

Another relevant anatomical factor identified in this study was the distance between the mandibular canal and the buccal cortex. Proximity of the mandibular canal to the buccal cortex has been recognized as a risk factor for unfavorable fractures, as this anatomical configuration reduces the amount of cancellous bone available to dissipate the forces applied during bone splitting [16,20]. In our study, the average distance in patients with non-conventional fracture patterns was 3.9 mm, consistent with the findings of Jiang et al. (2021), who reported that a distance of less than 4 mm between the mandibular canal and buccal cortex was significantly associated with unfavorable fracture risk during SSRO.

Although the distance from the mandibular lingula (ML) to the sigmoid notch and posterior ramus border did not show statistically significant associations with fracture pattern in our study, Smith et al. (1991) [19] concluded that performing horizontal osteotomies too high and too posterior relative to the ML significantly increases the risk of unfavorable fractures due to cortical fusion and lack of cancellous bone to absorb the splitting forces. Supporting this finding, De Souza Fernandes (2013) [20] concluded that the horizontal osteotomy should not be performed more than 4 mm above the X-line proposed in his study, which extends from the most posterior point of the anterior border of the mandibular ramus to the posterior border, parallel to the mandibular base.

Significant differences were observed between Type 1 and Types 3 (p = 0.000) and 4 (p = 0.000), as well as between Type 2 and Types 3 (p = 0.000) and 4 (p = 0.000). The absence of a statistically significant difference between Types 3 and 4 justifies the inclusion of both patterns in the same analytical group (case group), based on the similarity of findings (Table 3). This similarity supports the grouping of these patterns into a single category for statistical analysis.

We thus opted to classify fracture patterns into two groups according to their potential clinical consequences, recognizing that the less conventional and more undesirable patterns correspond to Type 4 fractures (Plooij, 2009). The rationale for distinguishing between conventional and non-conventional fractures lies in the fact that surgical experience and anatomical analysis more reliably predict the occurrence of Type 1 and 2 fracture patterns after SSRO. Moreover, mandibular rami classified as Types 3 and 4 showed similar biomechanical characteristics in terms of ramus thickness, which warrants further investigation into possible clinical repercussions, such as neurosensory outcomes in patients from each group.

The presence of third molars was not significantly associated with fracture pattern, consistent with the findings of Camargo et al. (2016) [15], who did not identify this variable as an independent risk factor for unfavorable fractures. This contrasts with studies suggesting that the presence of third molars increases the incidence of fractures [21], highlighting the need for further prospective studies to clarify this issue.

Our findings reinforce the importance of detailed preoperative assessment of mandibular anatomy to identify patients at higher risk for unfavorable fractures. The use of multislice computed tomography is essential for accurately measuring anatomical variables associated with this complication, enabling safer and more individualized surgical planning [22].

This study is subject to specific limitations inherent to its design, including potential selection bias and the absence of multivariate statistical analyses. For this reason, the generalizability of the findings should be proceeded with caution. It is also worth noting that some mandibular rami from both the case and control groups may have originated from the same patient, potentially introducing intra-individual correlation. Nevertheless, given that the pattern of BSSO separation is assumed to be independent on each operated side, the statistical analysis did not account for the potential dependence between paired observations, representing a possible methodological limitation. Furthermore, despite the use of reliable

software for the analysis of computed tomography scans, the use of automated measurement functions may have introduced some degree of bias into the results.

Moreover, although unfavorable or unexpected fracture patterns during SSRO may be influenced by mandibular anatomy, it is important to recognize that other factors may also be associated with atypical separation, including operator-dependent variables such as the surgical technique used, cutting instruments, and the surgeon's experience, which were not considered in this study.

The results of this study emphasize the importance of careful evaluation of mandibular anatomical factors in preventing unfavorable fractures during orthognathic surgery, since a reduced space between the buccal cortical bone and the mandibular canal, as well as decreased bone thickness at the level of the lingula, are associated with unconventional fracture patterns during SSRO. Preoperative identification of these features through computed tomography may contribute to individualized surgical planning and the prevention of complications. However, as this is an observational study, the findings should be interpreted with caution, and additional prospective studies are necessary to confirm these associations.

## Supporting information

**S1 Dataset. Data extracted during research.**
(PDF)

## Author contributions

**Conceptualization:** Tatiane Fonseca Faro, Allan Vinícius Martins-de-Barros, Fábio Andrey da Costa Araújo.

**Data curation:** Thalles Moreira Suassuna, Tatiane Fonseca Faro, Fernanda Souto Maior dos Santos Araújo, Fábio Andrey da Costa Araújo.

**Formal analysis:** Elenisa Glaucia Ferreira dos Santos, Sérgio Murilo Cordeiro de Melo Filho, Fábio Andrey da Costa Araújo.

**Investigation:** Thalles Moreira Suassuna, Elenisa Glaucia Ferreira dos Santos, Sérgio Murilo Cordeiro de Melo Filho, Carla Cecília Lira Pereira de Castro.

**Project administration:** Fábio Andrey da Costa Araújo.

**Supervision:** Tatiane Fonseca Faro, Allan Vinícius Martins-de-Barros, Fábio Andrey da Costa Araújo.

**Writing – original draft:** Thalles Moreira Suassuna, Sérgio Murilo Cordeiro de Melo Filho, Carla Cecília Lira Pereira de Castro.

**Writing – review & editing:** Fernanda Souto Maior dos Santos Araújo, Allan Vinícius Martins-de-Barros, Fábio Andrey da Costa Araújo.

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
