## [Decision Letter · Decision Letter 0]

26 Jun 2025

Dear Dr. Cecília Lira Pereira de Castro,

Thank you for submitting your manuscript to PLOS ONE. After careful consideration, we feel that it has merit but does not fully meet PLOS ONE’s publication criteria as it currently stands. Therefore, we invite you to submit a revised version of the manuscript that addresses the points raised during the review process.

**ACADEMIC EDITOR: In general, it is a well written manuscript with good concepts, but I would like to see more emphasis on the new aspects that the study brings to the literature. All the corrections must be done.**

We look forward to receiving your revised manuscript.

Kind regards,

Orion Haas Junior, DDS, OMFS, MSc, PhD

Academic Editor

PLOS ONE

Additional Editor Comments (if provided):

Reviewers' comments:

Reviewer's Responses to Questions

**Comments to the Author**

1. Is the manuscript technically sound, and do the data support the conclusions?

Reviewer #1: Yes

Reviewer #2: Yes

Reviewer #3: Yes

2. Has the statistical analysis been performed appropriately and rigorously?

Reviewer #1: Yes

Reviewer #2: Yes

Reviewer #3: Yes

3. Have the authors made all data underlying the findings in their manuscript fully available?

Reviewer #1: Yes

Reviewer #2: Yes

Reviewer #3: Yes

4. Is the manuscript presented in an intelligible fashion and written in standard English?

Reviewer #1: Yes

Reviewer #2: Yes

Reviewer #3: Yes

Reviewer #1: Dear Authors,

Thank you for the opportunity to review your manuscript. The topic addressed is both timely and relevant, especially considering its clinical implications in orthognathic surgery.

I have noted some points that may benefit from clarification or revision to strengthen the overall scientific rigor and clarity of the work. Please find below my detailed comments and suggestions for improvement.

1. Abstract

# It omits any mention of study limitations or statistical control for intra-individual observations.

Recommendation: Consider including a sentence noting the retrospective design and potential sources of bias.

2.Methods

#Add a brief paragraph discussing the sample size justification or power estimation

#Clarify whether bilateral rami were obtained from the same patients and, if so, apply appropriate statistical models to include the issue of intra-individual correlation, which is not accounted for in the statistical analysis.

3.Results

#The results section does not mention how missing data (if any) were handled.

#There is no clear statement about the number of patients versus number of rami, which may inflate the sample size and precision if not adjusted.

#The tables formatting does not adhere to standard scientific style. In scientific manuscripts, tables are usually presented without visible grid lines between rows and columns.

4.Discussion

#The authors do not address study limitations as (Retrospective design;Potential selection bias;Absence of adjustment for intra-individual correlation;No multivariate control of confounders),which is a critical omission in observational research.

#There is occasional language suggesting causality which should be revised to reflect associative relationships only.

5. Conlcusion

#Consider softening the conclusion by noting that the findings are associative and that further prospective studies are warranted.

6.Figures and tables

# Figures are helpful, though Figure 1 and 2 could benefit from improved resolution and clearer anatomical labels.

Consider to improve figure quality and consider labeling anatomical landmarks more explicitly for clarity.

#The tables formatting does not adhere to standard scientific style. In scientific manuscripts, tables are usually presented without visible grid lines between rows and columns. Please , consult PlOS One guidelines for table formating and presentation.

we sent a recommendation to the editor.

Reviewer #2: 1. Missing reference in introduction in second alinea explaining lingual split scale (ref 11).

2. please provide period in which these 390 cases were operated.

3. are the case operated upon consecutively or were the groups selected based on the type of fracture? The incidence of the type 1 and 2 versus 3 and 4 are approximately 60% versus 40%, which suggests that there would be more controles than cases. Please explain this more extensively.

Reviewer #3: Thank you for the invitation to review this manuscript. I have read it with great interest and found it to be an enjoyable read. I appreciate the effort put into this work and share the desire to see it published. I believe the authors have made a valuable contribution to the field of sagittal split ramus osteotomy (SSRO) postoperative outcomes.

Managing postoperative unfavorable fractures after SSRO represents one of the most formidable complications to address. Using preoperative & postoperative multislice computed tomography imaging, authors have provided a highly relevant insight to the risk factors associated with lingual fracture patterns in SSRO.

The manuscript is clearly written and well-structured and outcomes are well-chosen and relevant to the question of interest. The results and discussion are presented in a logical and accessible manner. The inclusion of 3rd molar presence & the distance from the mandibular lingula (ML) to the sigmoid notch and

posterior ramus border and its effect on surgery outcome is a strength and allows reader to have a more detailed understanding of the topic. I kindly request that the authors and reviewers take the following points into consideration prior to publication:

1- Can the authors include a priori power analysis to confirm that n=90 per group were sufficient?

2- The application of RadiAnt DICOM Viewer® for postoperative CT scans is commendable, though I wish authors would have added a brief section in addressing limitations of the automated functions.

3- Figure  2 is of insufficient quality. Kindly replace it with a higher-resolution version.

All the best to the authors for their further research.

**Do you want your identity to be public for this peer review?** For information about this choice, including consent withdrawal, please see our Privacy Policy

Reviewer #1: No

Reviewer #2: **Yes: ** J.M. Plooij, MD DMD PhD

Reviewer #3: No

---

## [Author Response · Author response to Decision Letter 1]

16 Jul 2025

We confirm that all relevant data are included in the manuscript and its Supporting Information files, and are referenced in the newly added 'Data Availability Statement' section.

---

## [Editor Report · Decision Letter 1]

18 Jul 2025

Anatomical Factors Associated with the Lingual Fracture Pattern in Sagittal Split Ramus Osteotomy: A Case-Control Study

PONE-D-25-23674R1

Dear Dr. Cecília Lira Pereira de Castro,

We’re pleased to inform you that your manuscript has been judged scientifically suitable for publication and will be formally accepted for publication once it meets all outstanding technical requirements.

Kind regards,

Orion Haas Junior, DDS, OMFS, MSc, PhD

Academic Editor

PLOS ONE
---

## [Editor Report · Acceptance letter]

PONE-D-25-23674R1

PLOS ONE

Dear Dr. Andrey da Costa Araújo,

I'm pleased to inform you that your manuscript has been deemed suitable for publication in PLOS ONE. Congratulations! Your manuscript is now being handed over to our production team.

Kind regards,

on behalf of

Dr. Orion Haas Junior

Academic Editor

PLOS ONE